# Communication is key: Mother-offspring signaling can affect behavioral responses and offspring survival in feral horses (*Equus caballus*)

**Cassandra M. V. Nuñez**[1,2,3¤]*, **Daniel I. Rubenstein**[3]

**1** Department of Biological Sciences, The University of Memphis, Memphis, Tennessee, United States of America, **2** Department of Natural Resource Ecology and Management, Iowa State University, Ames, Iowa, United States of America, **3** Department of Ecology and Evolutionary Biology, Princeton University, Princeton, New Jersey, United States of America

¤ Current address: Department of Biological Sciences, The University of Memphis, Memphis, Tennessee, United States of America

* cmnunez@memphis.edu

**Data Availability Statement:** Our data files are available from the Open Science Framework data repository at the following link: https://osf.io/tdeau/?view_only=4280ce9a274448a3bcaf0576278dff67

## Abstract

Acoustic signaling plays an important role in mother-offspring recognition and subsequent bond-formation. It remains unclear, however, if mothers and offspring use acoustic signaling in the same ways and for the same reasons throughout the juvenile stage, particularly after mutual recognition has been adequately established. Moreover, despite its critical role in mother-offspring bond formation, research explicitly linking mother-infant communication strategies to offspring survival are lacking. We examined the communicative patterns of mothers and offspring in the feral horse (*Equus caballus*) to better understand 1) the nature of mother-offspring communication throughout the first year of development; 2) the function(s) of mother- vs. offspring-initiated communication and; 3) the importance of mare and foal communication to offspring survival. We found that 1) mares and foals differ in when and how they initiate communication; 2) the outcomes of mare- vs. foal-initiated communication events consistently differ; and 3) the communicative patterns between mares and their foals can be important for offspring survival to one year of age. Moreover, given the importance of maternal activity to offspring behavior and subsequent survival, we submit that our data are uniquely positioned to address the long-debated question: do the behaviors exhibited during the juvenile stage (by both mothers and their young) confer delayed or immediate benefits to offspring? In summary, we aimed to better understand 1) the dynamics of mother-offspring communication, 2) whether mother-offspring communicative patterns were important to offspring survival, and 3) the implications of our research regarding the function of the mammalian juvenile stage. Our results demonstrate that we have achieved those aims.

**Funding:** This work was supported by the National Science Foundation (https://www.nsf.gov), IIS-0705311, to D.I.R. and by USDA National Institute of Food and Agriculture Hatch Multistate Funding (https://nifa.usda.gov/program/hatch-act-1887-multistate-research-fund), project IOW05509, to C. M.V.N. The funders had no role in study design, data collection and analysis, decision to publish, or preparation of the manuscript.

**Competing interests:** The authors have declared that no competing interests exist.

# Introduction

Acoustic signaling plays a critical role in mother-offspring recognition [1, 2] and by extension, successful bond-formation and offspring development [3]. However, we do not know if mothers and offspring signal acoustically in similar ways and for similar reasons throughout the juvenile stage, particularly after mutual recognition has been sufficiently established. Changes in vocal communication across ontogeny suggest that mothers and offspring are highly flexible in their use of acoustic signals, consistent with variation in signal function throughout the juvenile stage. For example, infant rodents will emit ultrasonic vocalizations when separated from their mothers; the occurrence of these calls increases during the first 6–7 days of life, peaking at day 8, and declining thereafter so that by 2 weeks of age, the calls have all but disappeared from the pup's repertoire [4]. Similarly, in domestic cats (*Felis silvestris catus*) mother-offspring vocal communication changes across the weaning process: with increasing development, separation signals are used less frequently and their fundamental frequency is reduced [5]. Such variation is clearly indicative of increasing offspring independence, but may also indicate changes in the reasons why mothers and offspring communicate.

Despite its critical role in mother-offspring bond formation, research explicitly linking mother-infant communication strategies to offspring development are lacking. Instead, researchers often manipulate weaning timing to alter mother-offspring bond formation and study the subsequent effects on offspring development. For example, research with mice (*Mus musculus*) shows that early weaned males exhibited long-lasting increases in anxiety and aggression, while early weaned females exhibited increased anxiety and lower maternal behavior in adulthood [6, 7]. These studies have provided significant insights, demonstrating important links between mother-offspring bond formation and offspring success, but only tangentially address the role of mother-offspring communication and do not at all address the ultimate survival of the offspring. Additional studies, specifically examining mother-offspring signaling and its potential impacts on offspring survival, are critical if we are to fully understand the significance of mother-offspring communication. In addition, understanding the communicative dynamics between mothers and their offspring could help shed light on the more general, long-standing question: what is the function of the juvenile stage: do the behaviors exhibited confer immediate and/or future benefits to young mammals? Typically, studies investigating the function of the juvenile stage have focused on offspring play behavior [8–12], essentially ignoring maternal behavior, a crucial factor affecting offspring behavior and subsequent survival. We examined the vocal communication patterns between mothers and their offspring in the feral horse (*Equus caballus*) throughout the first year of development to gain insights into 1) differences in mother- vs. offspring-initiated communication throughout the juvenile stage; 2) the function(s) of mother- vs. offspring-initiated communication throughout the juvenile stage; and 3) the importance of mare and foal communication to offspring survival to nutritional independence (one year of age in feral horses) [13, 14].

Feral horses are social ungulates, living in cohesive groups, or bands. Bands consist of usually one, but sometimes more than one male (stallion), one to several female(s) (mare(s)), and their offspring (foals) [15–17]. Bands are typically stable, with few changes in group composition over time [18]. Mares will often remain in the same group for most (if not all) of their adult lives [19]—allowing for persistent social bonds between them [20]—and offspring remain with the group until 4–6 years of age [14, 21]. Although the communicative repertoire of feral horses has been described as rudimentary [22], it has been shown to provide frequent and clear communication amongst group members [14, 15, 23, 24]. This natural history makes feral horses an excellent model species for the study of mother-offspring communication in the wild.

Here, we examined when, how, and why mares and foals engaged in acoustic signaling and assessed subsequent offspring survival. Specifically, we examined differences in the rates of vocal communication for mares vs. foals, the types of signals used by mares vs. foals to initiate communication, the context in which mares vs. foals initiated communication, and the subsequent outcomes of mare- vs. foal-initiated communication events during the first year of foal development to 1) better understand the role of mother- vs. offspring-initiated communication in mammals, 2) examine how variation in mother-offspring communication may contribute to offspring survival to nutritional independence [13, 14], and 3) gain some insight into the function of the juvenile stage in mammals.

We predicted that animals would use louder calls to ensure information transfer: 1) when initiating communication from farther distances, and; 2) when initiating communication in more closed habitats with lower visibility. Changes in acoustic signal production in similar contexts have been demonstrated in several taxa, including frogs [25], birds [26], primates [27], some whales [28], and insects [29]. We also predicted that with increasing foal age, 3) rates of mare-foal communication would decrease and that; 4) the outcomes of mare-foal communication would shift such that: a) decreases in mare-foal distance (approaches to one another) would decline in frequency; b) suckling behavior would decline in frequency; c) increases in mare-foal distance (leaves from one another) would increase in frequency; and d) no changes (to mare-foal distance or activity) would increase in frequency. As young mammals become less dependent upon their mothers, such patterns are common [5, 30, 31]. In addition, we predicted that 5) foals would be more likely to survive if both they and their mothers initiated communication at higher rates throughout the first year of development. Given the importance of mother-offspring recognition to successful bond formation [1, 2] and subsequent offspring development [3], we hypothesized that increased communication between mare and foal would build on this foundation, further strengthening the mother-offspring bond, and also ensure sufficient protection and nutrients to offspring when needed [32]. Finally, we predicted that 6) our data would support the hypothesis that the behavior demonstrated during the juvenile stage affords immediate (vs. future) benefits to offspring [33].

## Methods

### Study area

We studied a feral horse population living on Shackleford Banks, a barrier island approximately 3 km off the coast of North Carolina, USA. The island is protected land and is part of Cape Lookout National Seashore (https://www.nps.gov/calo/index.htm). Although Shackleford Banks was within the United States National Park Service's jurisdiction at the time of this study, the horses were largely unmanaged: there was no animal husbandry or animal care program. There were no predators of the horses in the study area. Shackelford is roughly 15 km in length, and ranges from 0.5 to 3 km in width, extending from approximately 34˚41'12.44 N, 76˚39'05.70 W to 34˚37'51.13" N, 76˚31'51.20" W. The Shackleford Banks horses, also known as Banker horses, are classified as Colonial Spanish horses; they are similar to South American Iberian breeds and are most closely related to the Venezuelan Criollo, Puerto Rican Paso Fino, and Marsh Tacky horses [34]. Their reproductive units are typical of feral equids, consisting of coherent bands of one or, sometimes two or three stallion(s) with one to several mare(s) and their offspring [17]. While multi-stallion bands constitute a large portion of some feral horse populations, (33% in the Kaimanawa horses, for example) [35], they occur less frequently on Shackleford Banks, accounting for only 19% of bands at the time of this study. In addition, differences in the degree of stallion territoriality have been associated with the island's varying

ecology [17]. During the time of the study, bands on the western portion of the island moved freely among largely overlapping home ranges (demonstrating 7.9% + 0.42% home range exclusivity), while bands on the mideastern and eastern portions of the island were more likely to defend territories (demonstrating 45.5% + 1.21% and 74.7% + 1.2% home range exclusivity, respectively. Regardless of island region, all bands remained spatially distinct from one another so that band membership was easily determined [15, 17, 36].

At the time of this study, membership in bands on Shackleford was long-lasting with most changes involving the dispersal of immature individuals (both male and female) at 4–6 years of age. Stallions sometimes fought to acquire mares from other groups, but almost always retained their mares [17, 37, 38]. Horses were identified individually by color, sex, age, physical condition, and other distinguishing markings including freeze brands. Ages were known from long-term records for the identified horses of Shackleford Banks [39].

## Study subjects

We observed 34 females and their 45 foals, constituting 63%, 65%, and 93% of the mare-foal pairs present in 1995, 1996, and 1997, respectively. Field logistics necessitated that we locate and observe animals on foot; this limited our ability to observe all mare-foal pairs present, especially in the first two years of the study. Study subjects were organized into 10 bands in 1995, 18 bands in 1996, and 11 bands in 1997 (see S1 Table). We monitored mare-foal pairs across the entire island: 38%, 23%, and 41% of the animals monitored were located in the western, mideastern, and eastern portions of the island, respectively. This was commensurate with the approximate area each island region comprised: 376, 126, and 308 hectares for the west, mideast, and east, respectively. Mare-foal pairs were monitored until foal nutritional independence at one year of age [13, 14]. Eighteen mares were studied for multiple years with their different foals (see S1 Table). Thirteen out of 45 foals (29%) died during the course of the study (6 of 15 in 1995, 6 of 17 in 1996, and 1 of 13 in 1997). Though cause of death was not routinely determined, during one examination, a female foal was found to be heavily infested with parasitic worms. At the time of this study, it was not policy of the National Park Service to track the mortality of individual animals.

## Sampling

We collected data from April 1995 to September 1997, totaling over 3000 hours of observation. Mares and foals were observed as dyads. We observed focal individuals (mares or foals) continuously for 15 minutes; during which time we recorded the activity of and distance to their counterpart every 5 minutes. Animals were observed from 5–10 meters away to ensure sufficient detection of mare-foal signaling. We remained still and silent during all sampling. Such distances can be disruptive for some animal populations; however, the Shackleford horses are largely habituated to human presence. At the time of this study, Shackleford Banks hosted over 100,000 visitors per year [40]. Upon our approach, the horses typically continued in their present activity or looked to the observer for some number of seconds before returning to their previous activity (personal observation, C.M.V.N).

Instances of communication between mare and foal were recorded *ad libitum* during both focal and scan sampling [41]. The signals of interest included the snort, nicker, and whinny (Table 1, Fig 1, S1–S3 Files). The snort appears to be a multi-faceted signal for both feral and domestic horses: it is used to maintain contact [42], as an alarm signal [23], during play [24], when an animal is "restless" [23], and may be indicative of positive emotions [43]. On Shackleford Banks, all horses, males, females, adults, and juveniles readily used snorts; snorts were softer relative to nickers and whinnies (Fig 1, S1 File); they were exchanged freely between all

**Table 1. The three communicative signals most commonly used by feral mares and their foals adapted from Yeon [24].**

| Signal | Duration (ms) | Fundamental frequency | Tonality | Context |
|---|---|---|---|---|
| Snort | 490–1,310 | - | Low | Clearing of nasal passages; general contact call for Shackleford horses |
| Nicker | 200–1,700 | 100–150 | Low-medium | Mother/young contact; recall; recognition |
| Whinny | 500–2,800 | 400–2,000 | High | Distress call; long-range communication |

band members, and were often used between mares and their foals (personal observation, C. M.V.N). This behavior seemed most similar to what has been described in domestic horses

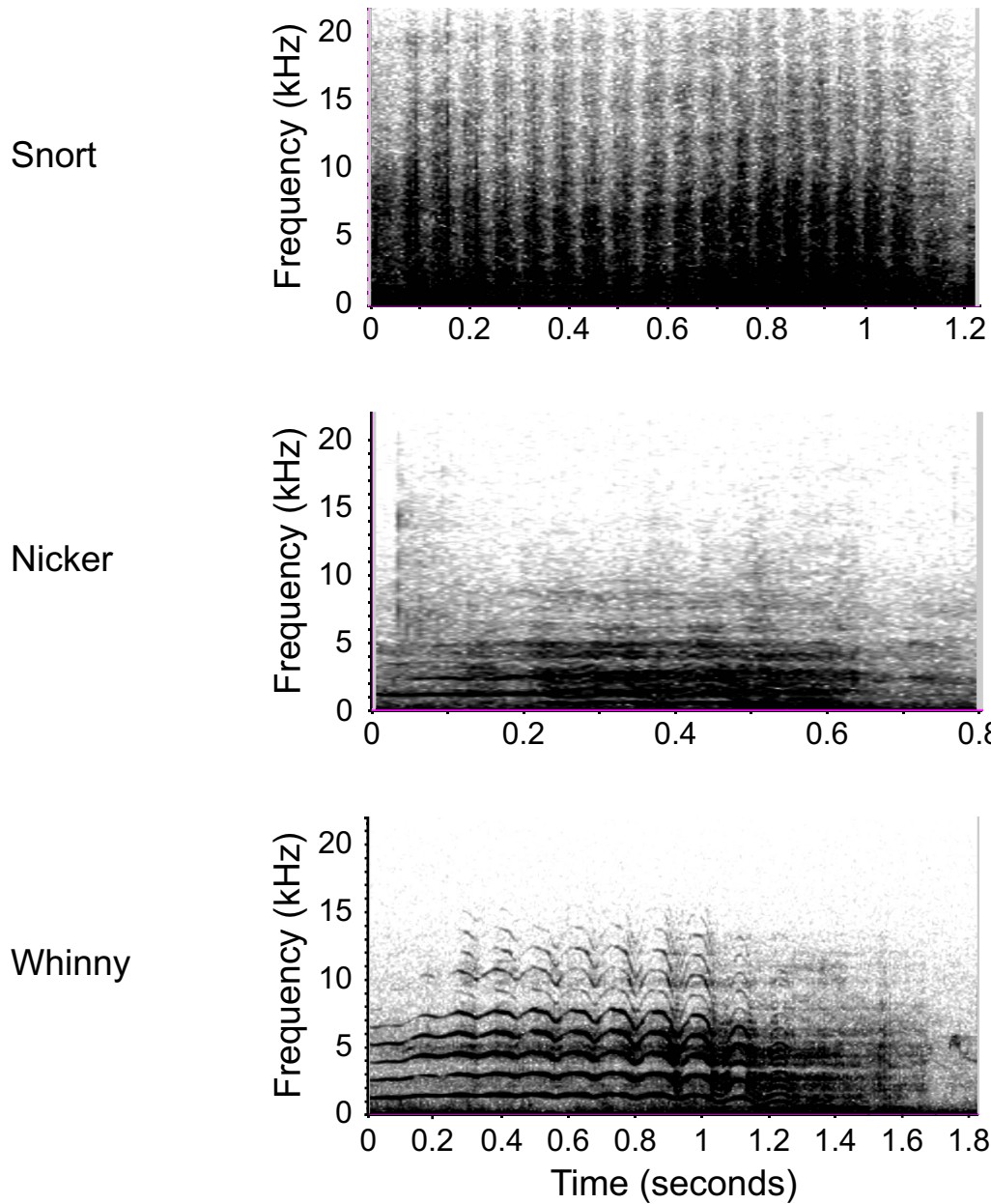

**Fig 1. Spectrograms of adult mares using the three calls examined in this study.**

whereby animals appeared to use snorts to communicate their position to the group [42]. Given the apparent diverse nature of this signal, we do not mean to define it overly strictly; however, between mother and offspring in particular, our observations suggest that the snort likely served as a contact call. Nicker and whinny use by mares and foals more closely resembled what has been described in other feral horse populations [15, 22].

We examined differences in the signals mares and foals used to initiate communication. We recorded 956 communication events during the first year of foal development; 470 (49%) were mare-initiated, 486 (51%) were foal-initiated. Communication events occurring ≥ 2 minutes apart were considered new events, i.e. not responses to previous signaling. Snorts ($n = 522$), nickers ($n = 96$), and whinnies ($n = 315$) constituted 55%, 10%, and 33% of all signals, respectively (see Table 2 for additional details). The remaining 2% of signals ($n = 22$) were categorized as "whinny/nickers" (a combination of the two sounds) or "other". Communication events initiated with these signals were excluded from the analyses as they constituted such a small fraction of the dataset.

Activities other than communication recorded during sampling included feeding, suckling, resting, walking, play, and self- and allo-grooming. To estimate mare-foal distance in meters we counted the number of horse lengths between individuals and multiplied this measure by 2 (Shackleford horses are ∼2 m long). To determine the effect of habitat characteristics on mare-foal communication events, we used a variation of the Bitterlich method [44] to measure the density of trees and shrubs that likely obstructed visibility. For these measurements, the investigator's arm was held straight in front of the body at eye level; all trees and shrubs that appeared larger than the observer's fist were counted as the observer turned 360˚. This sum was then divided by the total number of obstructions in the area to produce a habitat's score, ranging from 0–1. Habitats with a score between 0 and 0.4 were categorized as having high visibility; those with a score between 0.5 and 1 were categorized as having low visibility.

For each communication event, we recorded the following:

- Initiator of the communication event (mare or foal)

- Signal used to initiate the communication event (snort, nicker, or whinny)

- Subsequent outcome of the communication event (no change to mare-foal activity or distance, decreases in mare-foal distance, increases in mare-foal distance, or suckling behavior); outcomes were considered related to the communication event if they occurred within 1 minute of the event

- Visibility of the area in which the communication event was initiated

Finally, physical condition is an important factor to consider as mothers in better condition will be more able to provide nutrients and likely, protection to their foals. We assessed mare condition via rump scoring. We determined rump scores examining the curvature of the line

**Table 2. Overall signal use by mares and foals during the first year of foal development.**

| Signal | Mare | Foal |
|---|---|---|
| Snort | 74.2% ($n = 349$) | 35.6% ($n = 173$) |
| Nicker | 9% ($n = 43$) | 10.9% ($n = 53$) |
| Whinny | 15.5% ($n = 73$) | 49.8% ($n = 242$) |

We analyzed 934 communication events. Communication initiation rates by mares and foals were calculated as follows: Number of communication events initiated (by mare or foal)/Total hours observed per age range.

between the tailbone and the point of the hip. Scores were based on a scale from 1 to 5; a score of 1 being the poorest [45].

## Statistical analyses

We analyzed the individual communication events of mare-foal pairs in R version 3.3.0 [46]. To test for differences in the rates at which mares and foals initiated communication (defined as the number of initiations per hours observed), we used linear mixed effects models in the nlme package [47]. Rates were log-transformed to achieve normality of model residuals and meet model assumptions. We used generalized linear mixed effects models with a binomial distribution and the glmerControl optimizer function in the lme4 package [48] to investigate the following: 1) differences in mare vs. foal signal use (snort, nicker, or whinny); 2) differences in the outcomes (no change to mare-foal distance or activity, mare-foal distance decrease, or suckling behavior) of mare- vs. foal-initiated communication; and 3) the probability that communication was initiated by a mare vs. a foal. Incidences of increased mare-foal distance following communication events were low, comprising only 3.2% ($n = 30$) of all recorded outcomes and were not considered in the final analyses.

To determine the drivers behind signal choice and the communication rates of mares and their foals, we included initial mare-foal distance, habitat visibility, mare age, foal age, and foal sex as fixed effects. Except when examining the factors affecting whether mares or foals initiated communication, we investigated possible interactions between all fixed effects and the initiator of communication (mare vs. foal). For each initial model (including all main effects and interactions), we used Akaike's Information Criterion adjusted for small sample size [49] to test model fits with 1) mare ID as a random effect, 2) foal ID as a random effect, or 3) mare and foal ID as nested random effects. For most models, using mare and foal ID as nested random effects improved model fit. To test for effects of individual band membership, we used AICc analysis to test model fits with and without the appropriate ID variable nested within band. For 55% of the models (5 out of 9), including band improved model fit (see S2 Table).

Not surprisingly, mare age was correlated with parity (Linear Mixed Effects Modal: estimate = -3.56, SE = 1.22, t = -2.93; $P = 0.01$); given that mare age encompassed both parity and presumably a host of other traits associated with increased experience, we elected to include mare age vs. parity in all of our models. Similarly, island region (east, mideast, and west [17]) and band social system (territorial vs. non-territorial [17]) were associated with habitat visibility (Linear Mixed Effects Model: region and habitat visibility: west vs. east, $P < 0.01$; west vs. mideast, $P = 0.01$; east vs. mideast, $P = 0.50$; social system and habitat visibility, $P < 0.0001$). As habitat visibility is likely the more relevant factor regarding mare-foal signaling [14, 22, 27], we did not include island region or band social system in our models. Models were simplified using backwards elimination, retaining all main effects and removing interaction terms with $P$-values $> 0.05$. Effects were considered significant if $P \leq 0.05$.

To determine how foal survival to nutritional independence varied with rates of mare- and foal-initiated communication, we performed survival analyses with parametric accelerated failure time models assuming a Weibull distribution using the "survreg" function in the "survival" package (R package version 3.3.2) [50] in R version 3.3.0 [46]. These models allow for the inclusion of continuous variables and the Weibull distribution makes fewer assumptions about the risk of death over time than do other distributions [51]. Models included both mare and foal initiation rate, average foal suckling rates, average distance maintained between mare and foal, average mare condition, mare age, and foal sex. We used linear models to determine whether communication rates predicted average suckling rate, average mare-foal distance, or average mare condition. Because the earliest foal death occurred at 20 weeks of age, we only

considered acoustic signaling occurring during the first 20 weeks of development (from the same 34 mares and their 45 foals, totaling 735 communication events) for these analyses. Specifically, our survival models included separate predictor variables of communication rates from 0–10 weeks and 10–20 weeks. All databases can be accessed from the Open Science Framework data repository at https://osf.io/tdeau/?view_only=4280ce9a274448a3bcaf0576278dff67.

## Ethical note

All sampling was conducted in accordance with ASAB/ABS Guidelines for the Use of Animals in Research [52]. Given the non-invasive, strictly observational nature of this study, neither the Princeton University nor the National Park Service Institutional Animal Care and Use Committees deemed permitting necessary. Moreover, at the time of this study, the National Park Service at Cape Lookout National Seashore did not require research permits from our research group.

During our observations, we remained still and silent and conducted all data collection from 5–10 meters away; this distance ensured that we did not disturb the animals and yet was near enough to facilitate the detection of mare-foal signaling. Though such distances can be disruptive for some animal populations, the Shackleford horses are largely habituated to human presence. At the time of this study, Shackleford Banks hosted over 100,000 visitors per year [40]. Upon our approach, individuals typically continued in their present activity or looked to the observer for some number of seconds before returning to their previous activity (personal observation, C.M.V.N). Shackleford Banks is protected land and is currently managed by the United States National Park Service (https://www.nps.gov/calo/index.htm) and the Foundation for Shackleford Horses (https://www.shackleford-horses.org). At the time of the study, the horses were largely unmanaged: although the island was within the National Park Service's jurisdiction, there was no animal husbandry or animal care program. Aside from the observational sampling of the horses described above, no protected species were sampled in this study.

## Results

### Communication initiation

During the first year of development, mares and foals initiated communication at similar rates (Linear Mixed Effects Model: estimate = -0.004, SE = 0.03, $t$ = -0.13, $P$ = 0.90; mare mean rate = 0.497 initiations/hour, SE = 0.04; foal mean rate = 0.499 initiations/hour, SE = 0.04) and there was a positive correlation between mare- and foal-initiated communication (Linear Mixed Effects Model: estimate = 0.35, SE = 0.04, $t$ = 8.57, $P$ < 0.00001). There was no effect of foal sex, foal age, or mare age on the rates at which mares and foals initiated communication (all $P$-values $\geq$ 0.17, S3 Table).

Mares and foals differed in the average distance at which they initiated communication and in the signals they used most often. Mares initiated communication at shorter mare-foal distances, than did their foals (Generalized Linear Mixed Effects Model, initiator type (mare): estimate = -0.04, SE = 0.007, $t$ = -5.60, $P$ < 0.00001, Fig 2) and were more likely to use snorts, the softest form of communication (Generalized Linear Mixed Effects Model: estimate = 2.55, SE = 0.45, $z$ = 5.64, $P$ < 0.00001), than foals were. Foals used louder calls more frequently than their mothers did. For example, foals were more likely to use whinnies (Generalized Linear Mixed Effects Model, initiator type (mare): estimate = -3.27, SE = 0.59, $z$ = -5.54, $P$ < 0.00001) and, to a lesser extent, nickers (Generalized Linear Mixed Effects Model: initiator type (mare); estimate = -1.27, SE = 0.39, $z$ = -3.25, $P$ = 0.001) than mares were. In addition, there was an

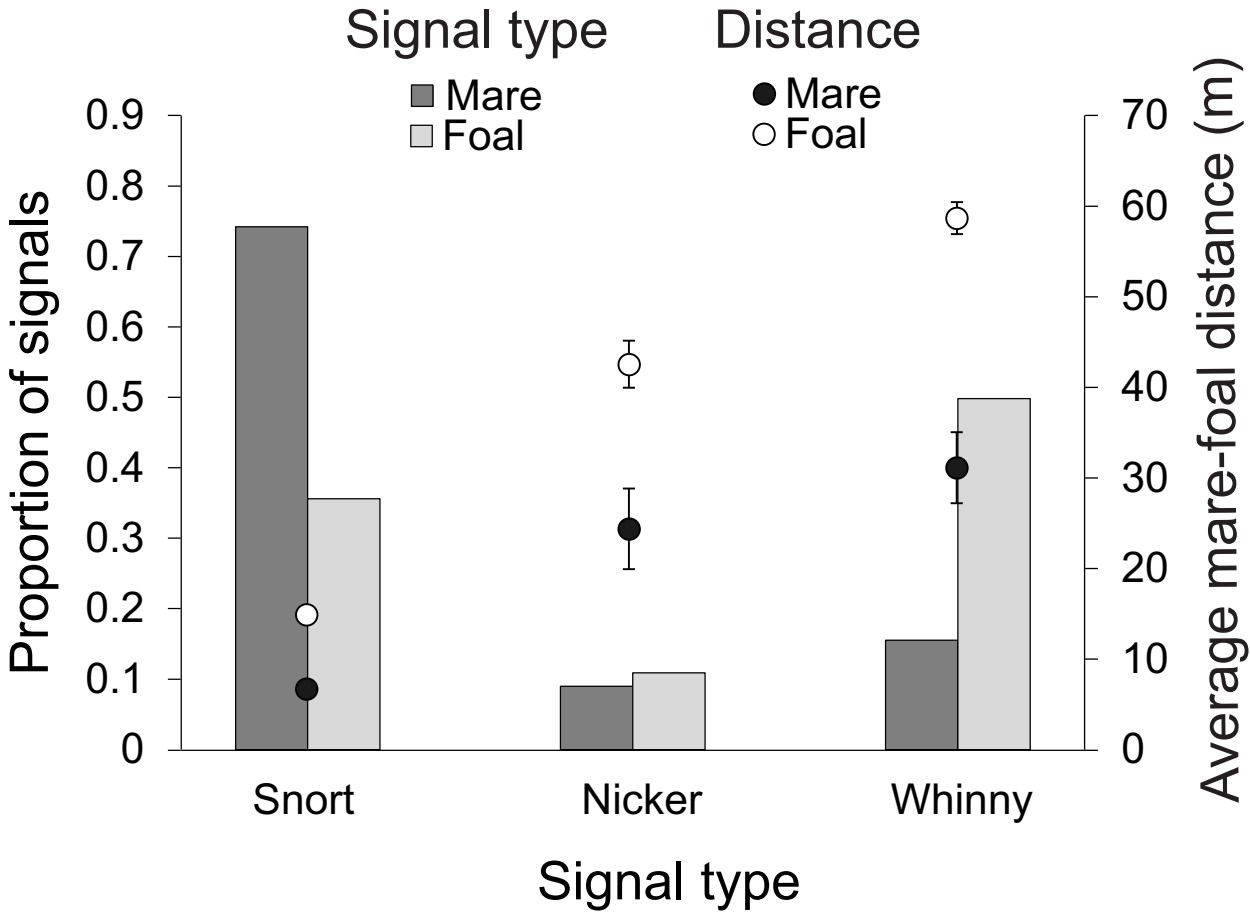

**Fig 2. The proportion of signals used by mares and foals and the average distances at which they were more likely to use them.** On average, mares were more likely to initiate communication at closer distances and were more likely to use snorts than their foals were. Foals were more likely to initiate communication at farther distances and were more likely to use nickers and whinnies than their mothers were. Error bars show ± 1 standard error. Standard error values for the average distances at which mares and foals used snorts are low enough that they are obscured by the points.

interaction between initiator type and foal age: mares seemed to play a more active role with their foals aged $\geq$ 25 weeks, initiating communication more often than their foals did (Generalized Linear Mixed Effects Model, initiator type (mare) x foal age: estimate = 0.02, SE = 0.009, $z$ = 2.36, $P$ = 0.02). Habitat visibility, foal sex, and mare age did not affect whether mares or foals initiated communication (all $P$-values $\geq$ 0.29, S3 Table).

### Signal use

**Snorts.** Like their mothers, foals used snorts when initiating communication from closer distances (Generalized Linear Mixed Effects Model: estimate = -0.12, SE = 0.01, $z$ = -9.56, $P$ < 0.00001). There was an interaction with initiator type and foal age: with increasing development, the probability that mares would initiate communication with snorts increased only moderately (Generalized Linear Mixed Effects Model, initiator type (mare) x foal age: estimate = -0.05 SE = 0.02, $z$ = -2.03, $P$ = 0.04, Fig 3A), while there was a marked increase in the probability that foals would initiate communication with snorts (Generalized Linear Mixed Effects Model: estimate = 0.07, SE = 0.02, $z$ = 3.96, $P$ < 0.00001, Fig 3B). In addition, while mares were less likely to use snorts in areas of low visibility (Generalized Linear Mixed Effects Model,

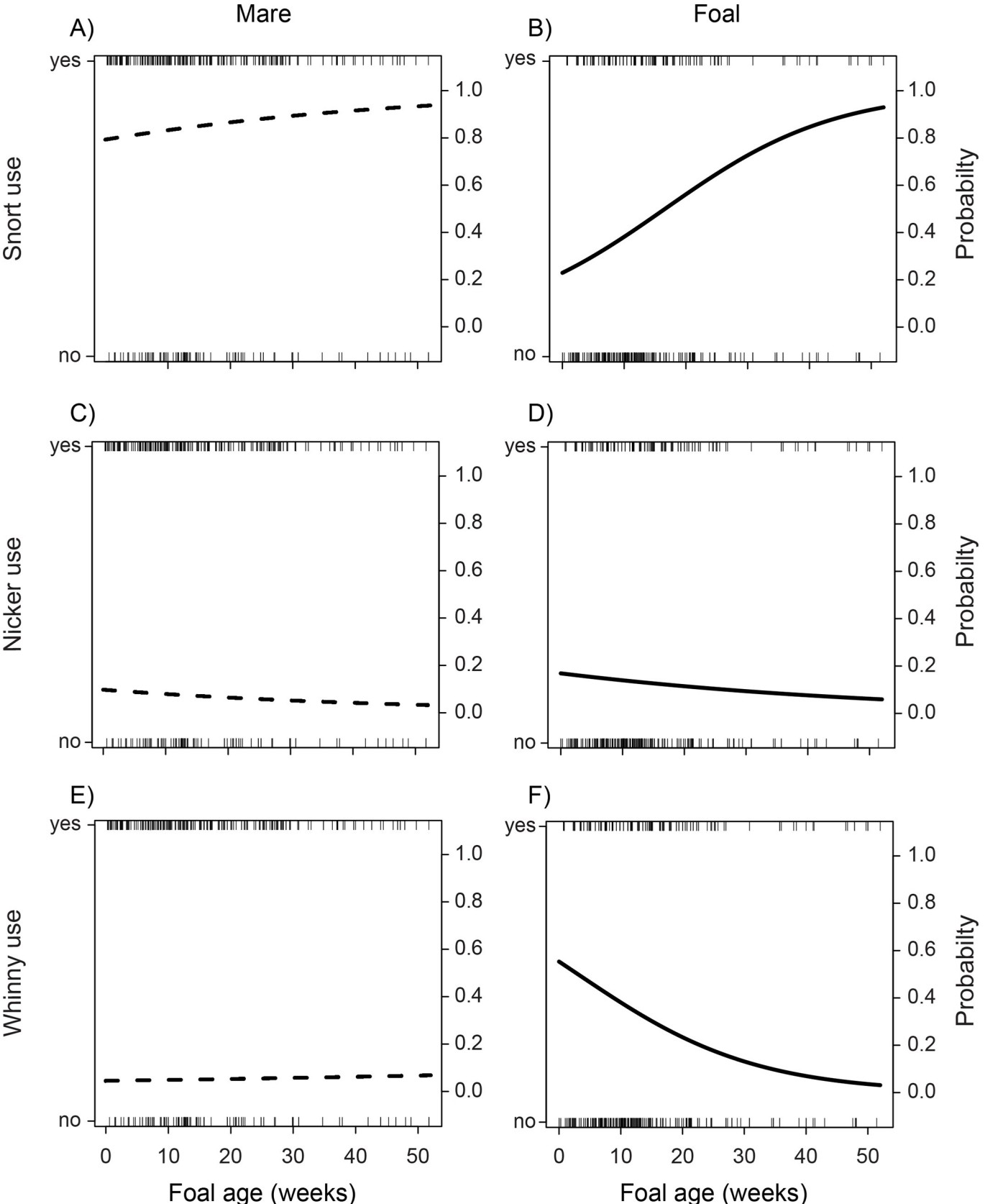

**Fig 3.** Changes with foal age in the use of snorts (A and B), nickers (C and D), and whinnies (E and F) by mares and their foals. While mares were largely consistent with their use of signals throughout foal development, foals showed more significant changes in their use of different signals (especially snorts and whinnies) over time. "Yes" and "No" categories indicate that a signal other than the one depicted was used to initiate communication. Segments show individual communication events initiated by mares and foals; dotted and black lines represent model predictions for mares and foals, respectively.

initiator type (mare) x visibility: estimate = -1.33, SE = 0.50, $z$ = -2.64, $P$ = 0.008), habitat visibility did not seem to affect whether or not foals used snorts (Generalized Linear Mixed Effects Model: estimate = 0.55, SE = 0.35, $z$ = 1.60, $P$ = 0.12). Interactions with initiator type and initial mare-foal distance, foal sex, and mare age proved not significant (all $P$-values > 0.70) and were removed from the final model. There was no effect of foal sex or mare age on the probability of snort use by mares or foals (all $P$-values > 0.60, S3 Table).

**Nickers.** Initial mare-foal distance was correlated with mare, but not foal, nicker use. The probability that mares would use nickers to initiate communication was increased at closer distances (Generalized Linear Mixed Effects Model, initiator type (mare) x mare-foal distance: estimate = 0.05, SE = 0.01, $z$ = 3.11, $P$ = 0.002), but the probability that foals used nickers did not seem affected by the initial mare-foal distance (Generalized Linear Mixed Effects Model: estimate = 0.006, SE = 0.01, $z$ = 0.64, $P$ = 0.52). Interactions with initiator type and habitat visibility, foal sex, foal age, and mare age proved not significant (all $P$-values > 0.10) and were removed from the final model. There was no effect of habitat visibility, foal sex, foal age, or mare age on the probability of nicker use by mares or foals (all $P$-values > 0.10, Fig 3C and 3D and S3 Table).

**Whinnies.** While the probability that mares used whinnies to initiate communication remained at a more or less constant rate throughout foal development (Generalized Linear Mixed Effects Model: estimate = 0.08, SE = 0.03, $z$ = 2.91, $P$ = 0.003, Fig 3E), the probability that foals used whinnies to initiate communication decreased with foal age (Generalized Linear Mixed Effects Model: estimate = -0.07, SE = 0.02, $z$ = -3.35, $P$ = 0.0008, Fig 3F). In addition, there was an interaction with habitat visibility whereby mares increased their use of whinnies in low visibility areas (Generalized Linear Mixed Effects Model, initiator type (mare) x visibility: estimate = 1.78, SE = 0.60, $z$ = 2.94, $P$ = 0.003), while foals did not (Generalized Linear Mixed Effects Model: estimate = -0.67, SE = 0.39, $z$ = -1.73, $P$ = 0.08). Interactions with initiator type and initial mare-foal distance, foal sex, and mare age proved not significant (all $P$-values > 0.10) and were removed from the final model. There was no effect of foal sex or mare age on the probability of whinny use by either mares or their foals (all $P$ values > 0.30, S3 Table).

## Signal use and communication outcomes

Overall, foal-initiated communication was more likely to result in suckling or a decrease in mare-foal distance than was mare-initiated communication (Generalized Linear Mixed Effects Model, (initiator type (mare)): estimate = -0.55, SE = 0.23, $z$ = -2.13, $P$ = 0.03) and there was a correlation with foal age whereby suckling behavior and/or decreases in mare-foal distance were less likely to occur with increasing foal development (Generalized Linear Mixed Effects Model: estimate = -0.02, SE = 0.008, $z$ = -2.43, $P$ = 0.01, Fig 4). Decreases in mare-foal distances averaged 12.10 meters (range = 0.16–96.00 meters).

Communication events initiated with whinnies were more likely to result in suckling behavior and/or a decrease in mare-foal distance than were events initiated with snorts, regardless of initiator type (Generalized Linear Mixed Effects Model: estimate = 1.41, SE = 0.30, $z$ = 4.75, $P$ < 0.00001; (initiator type (mare)): estimate = -0.26, SE = 0.42, $z$ = -0.61, $P$ = 0.54). On the other hand, communication events initiated with nickers were more likely to

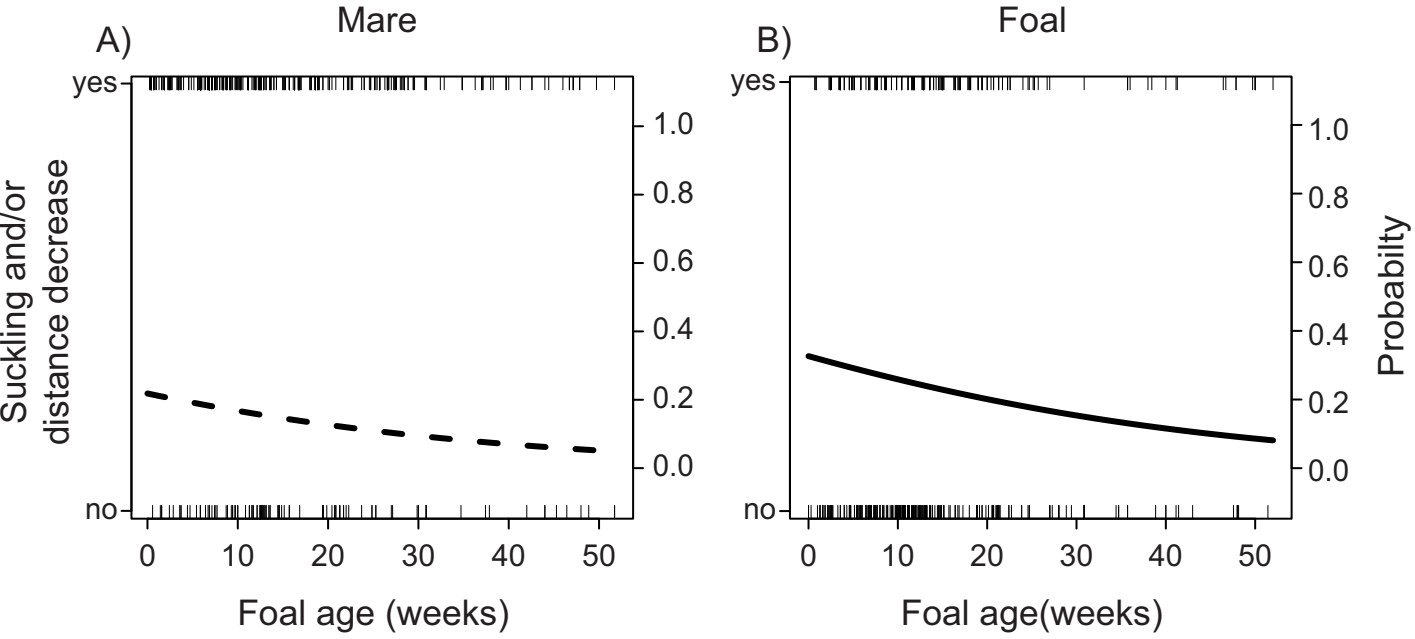

**Fig 4.** Changes with foal age and communication outcome by (A) mares and (B) foals. The likelihood that communication resulted in either suckling behavior or a decrease in mare-foal distance decreased with increasing foal age, regardless of initiator type (mare or foal). Segments show individual communication events initiated by mares and foals; dotted and black lines represent model predictions for mares and foals, respectively.

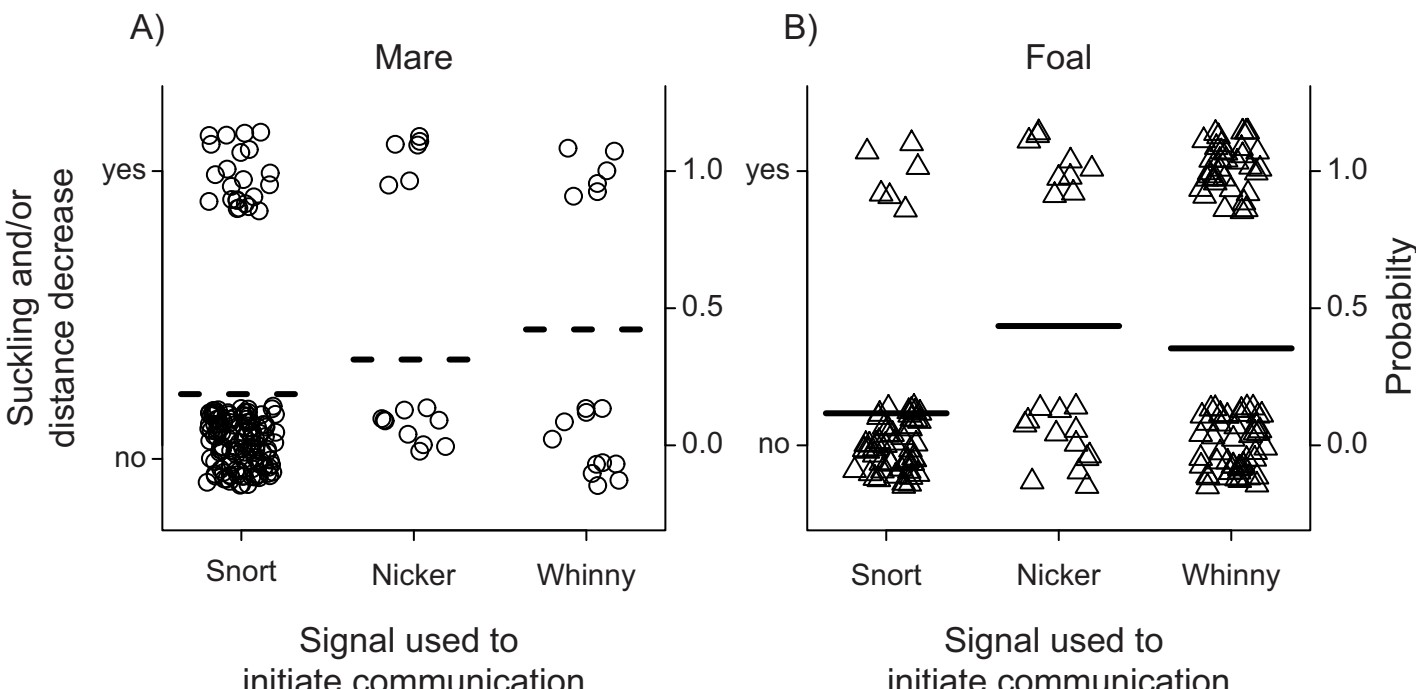

**Fig 5.** Signal use and outcome by (A) mares and (B) foals. Communication events initiated with whinnies were more likely to result in suckling behavior and/or a decrease in mare-foal distance than were those initiated with snorts, regardless of initiator type. Communication events initiated with nickers were more likely to result in suckling or decreases to mare-foal distance when used by foals, but not mares. Outcomes occurring from 0–10 weeks of age are shown. Circles and dotted segments represent individual communication events initiated by mares and model predictions, respectively; triangles and black segments represent individual communication events initiated by foals and model predictions, respectively.

result in suckling or decreases to mare-foal distance when used by foals, but not mares (Generalized Linear Mixed Effects Model: estimate = 1.76, SE = 0.40, $z$ = 4.52, $P$ < 0.00001; (initiator type (mare)): estimate = -1.08, SE = 0.54, $z$ = -1.98, $P$ = 0.05). When mares used nickers to initiate communication, the outcome was more likely to be no change to the animals' distance or current activity. These effects were especially evident during the first 10 weeks of development (Fig 5). Models separately estimating the effects of initiator type and signal use on the likelihood of suckling and distance decrease outcomes yielded equivalent results (S4 Table).

### Communication rate and foal survival

Even when controlling for average suckling rate, average mare-foal distance, and average mare condition, the rates at which mares initiated communication with their foals were correlated with offspring survival (Fig 6A). Foals with mothers that initiated communication at higher rates during the first ten weeks of development survived longer than did foals with mothers that initiated communication at lower rates (Survreg Model, mare communication rate: estimate = 1.25, SE = 0.55, $z$ = 2.30, $P$ = 0.02; mare average condition: estimate = 0.30, SE = 0.12, $z$ = 2.34, $P$ = 0.02). Mare communication rate did not predict average suckling rate (Linear Model: estimate = 0.18, SE = 0.15, $t$ = 1.24, $P$ = 0.22), average mare-foal distance (Linear Model: estimate = 0.28, SE = 0.40, $t$ = 0.71, $P$ = 0.50), or average mare condition (Linear Model: estimate = 0.02, SE = 0.06, $t$ = 0.40, $P$ = 0.70). The rates at which foals initiated communication did not appear to affect offspring survival (Survreg Model: estimate = -0.01, SE = 0.22, $z$ = -0.06, $P$ = 0.94, Fig 6B).

Rates of mare- and foal-initiated communication occurring during 10–20 weeks of age were not correlated with foal survival (Survreg Model, mare: estimate = -0.04, SE = 0.08, $z$ = -0.50, $P$ = 0.62; foal: estimate = 0.03, SE = 0.23, $z$ = 0.13, $P$ = 0.90), though correlations with foal survival and mare average condition remained significant (Survreg Model, mare average condition: estimate = 0.22, SE = 0.07, $z$ = 2.95, $P$ = 0.003). Average suckling rate, average mare-foal distance, foal sex, and mare age did not predict foal survival to nutritional independence in our models for foals aged 0–10 weeks (all $P$-values > 0.20, S3 Table). Similarly, average suckling rate, average mare-foal distance, and mare age did not predict foal survival to nutritional independence in our models for foals aged 10–20 weeks (all $P$-values > 0.40, S3 Table); however, male foals reaching this age seemed to have a higher chance of survival than females (Survreg Model, foal sex (male): estimate = 0.29, SE = 0.14, $z$ = 2.04, $P$ = 0.04). Variation in average suckling rate and mare-foal distance between foals that survived and those that did not was low (Table 3), indicating strong selection on these factors. Because the earliest foal death occurred at 20 weeks of age, we analyzed communication data from foals aged 0–20 weeks only.

## Discussion

Here, we show that in the feral horse (*E. caballus*), 1) important variation in mother-infant communication persists at least until nutritional independence; 2) this variation influences the outcomes of communication events; and 3) mare communication initiation rate is associated with offspring survival. To the best of our knowledge, ours is the first study to attempt an explicit link between mother-offspring communicative patterns and offspring survival in mammals. Our results contribute to a broader understanding of the nature and importance of mother-offspring communication.

During the first year of development, mares and foals differed in how and when they initiated communication. On average, mares initiated communication at closer mare-foal distances and preferentially used snorts (Fig 2), the quieter form of communication, while foals

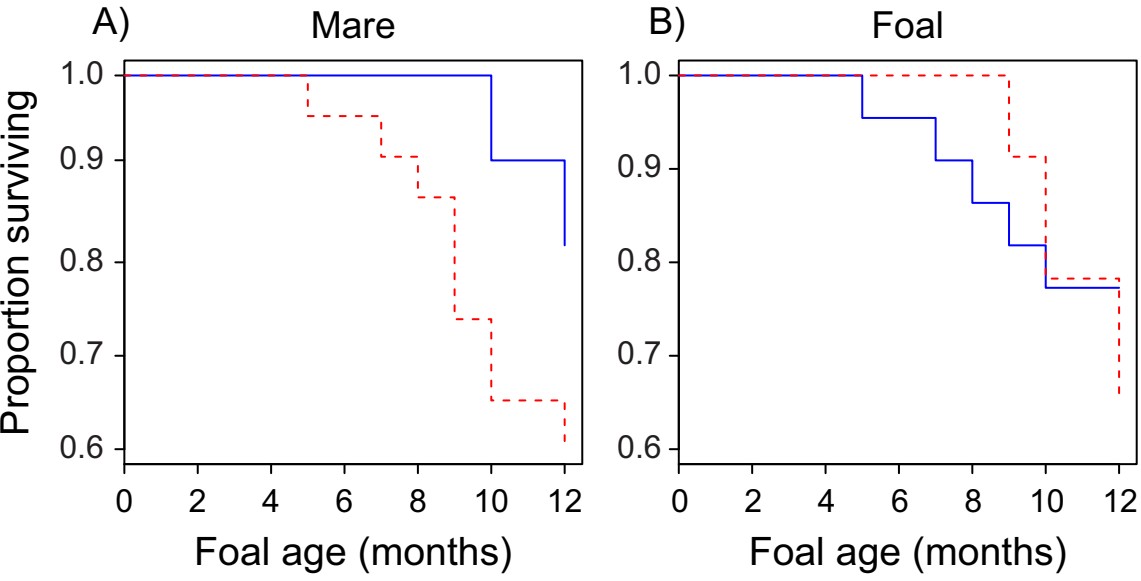

**Fig 6.** Communication initiation rate (number of mare or foal initiations/total hours observed per age range) by (A) mares and (B) foals and foal survival to nutritional independence (1 year of age). Foals with mares that initiated communication at higher rates (greater than the median value, solid blue line) during the first 10 weeks of foal development were more likely to survive than were foals with mares that initiated communication at lower rates (less than the median value, dotted red line). We did not detect an effect of foal initiation rate on foals' own survival.

initiated communication at farther distances and preferred whinnies and, to a lesser extent, nickers (Fig 2), the louder forms of communication. Moreover, the outcomes of mare- vs. foal-initiated communication differed: when mares initiated, the outcome was more likely to be no change to the animals' activity or distance; conversely, when foals initiated, suckling and/or decreases to the mare-foal distance were more likely to occur (Fig 4). Taken together, these results indicate an important difference in the function of mare- vs. foal-initiated communication. Mares seemed to use communication primarily as a means of balancing maternal attentiveness and foal independence [32]. Mares that maintain auditory contact with their foals, even when they are close by, consistently make their position known to the offspring which may enable foals to stray farther, enabling more exploration of the environment and socialization with fellow group members, two factors likely critical to foal survival [32, 53, 54]. On the other hand, foals seemed to use communication largely as a way of acquiring proximity to and/or nutrients from their mothers: indicating that for foals, communication functioned to ensure safety and nutritional resources.

Importantly, the specific signals mares and foals used to initiate communication were associated with the outcomes of communication events: whinnies were more likely than snorts to result in suckling behavior and/or decreases to mare-foal distance, regardless of whether

**Table 3. Summary statistics for average suckling rate and average mare-foal distance for foals aged 0–10 weeks that do and do not survive to nutritional independence.**

| Variable | Foal survival status | Mean | Median | Range | Inter-quartile range |
|---|---|---|---|---|---|
| Average suckling rate | Live | 2.01 | 1.80 | 0.56–6.09 | 1.29–2.53 |
| | Die | 1.70 | 1.93 | 0.86–2.80 | 1.20–2.20 |
| Average mare-foal distance | Live | 9.54 | 7.06 | 0.33–35.00 | 4.23–13.51 |
| | Die | 9.24 | 7.70 | 0.73–25.70 | 3.90–11.50 |

mares or foals initiated communication (Fig 5). On the other hand, nickers seemed to elicit different outcomes depending on initiator type: while they were more likely to result in no change to the animals' distance or current activity when used by mares to initiate communication, they were more likely to result in suckling and/or decreases in the mare-foal distance when used by foals (Fig 5). The use of nickers by mares and foals in lieu of the signals they more commonly use (snorts and whinnies, respectively) may be indicative of situational "urgency" or the level of mare and foal arousal [55–58]. For example, the use of nickers vs. snorts by mares may indicate increased arousal: if snorts are not effective, mares may use the louder nicker to ensure contact is made with their foals. Conversely, the use of nickers vs. whinnies by foals may indicate decreased arousal: if foals are closer to their mothers, more aware of her position, and/or not as hungry, foals may opt to use nickers in lieu of whinnies to acquire maternal resources. Still, whinnies and snorts seem to convey similar information, whether they are used by mares or foals. Considering the frequency with which mares and foals use these signals, our results suggest that preferential use of different calls can enable the seemingly varying functions of mare- vs. foal-initiated communication. Mares use softer vocalizations more frequently, which are unlikely to elicit suckling or changes in distance, but may serve to maintain contact. On the other hand, foals, and particularly young foals, in need of nutrition and protection from their mothers, are more likely to use nickers and whinnies, signals that more often elicit suckling or decreased mare-foal distance.

The fact that several aspects of mare- vs. foal-initiated communication were associated with increasing foal age provides evidence that the function(s) of mother-offspring communication changed with increasing foal independence, thus supporting our hypothesis. As offspring become more autonomous, their need for maternal proximity and milk decreases [5, 14, 30, 31]. As such, the function of offspring communication may converge with that of their mothers, shifting from resource acquisition to contact maintenance. For example, with increasing foal age: 1) foals became more likely to use snorts in lieu of whinnies, and; 2) suckling and/or decreases to the mare-foal distance became less likely outcomes of communication. Moreover, as foals aged, the probability that mares would initiate communication increased, but they remained more or less constant in their signal choice. It would seem, therefore, that mares augmented their efforts at keeping in contact with their increasingly independent foals. Still, increased mare attentiveness may have facilitated foal independence in that foals could explore their social and physical environment while also maintaining maternal contact should protection and/or nutrients become necessary [32]. The fact that we did not detect an effect of mare age (a proxy for maternal experience) on their communicative strategy is surprising. However, the signal repertoire of feral horses is fairly rudimentary; Feh [22] described it as "remarkably poor" given the species' level of sociality. The simplistic nature of feral horse signaling may limit the effects that mare experience can have on its execution.

Mares and foals initiated communication with each other at similar rates across development, perhaps indicating a mutual feedback between the two, such that more/less communicative behavior in mares induces similar behavior from their foals and vice versa. Perhaps more interesting, is the fact that mare communication initiation rate early in offspring development was associated with future foal survival to nutritional independence. Even when controlling for average suckling rate, average mare-foal distance, and average mare condition, foals were more likely to survive to one year of age if their mothers initiated communication at higher rates during the first 10 weeks of development (Fig 6). Contrary to our prediction, this effect did not hold for older foals (those 10–20 weeks of age), suggesting a critical period during which communication between mare and foal was of greater importance. This is unlike the apparent effect of mare condition, which was associated with foal survival for both 0–10 and 10–20 week-old foals. Given the nature of mare-initiated communication, it seems that

keeping contact with the foal during early development could be an essential factor in increasing the likelihood of foal survival.

It is somewhat surprising that the same cannot be said of foal-initiated communication during this period: contrary to our prediction, the rates at which foals initiated communication with their mothers did not correlate with their chances of future survival (Fig 6). Given that suckling behavior and decreases to mare-foal distance were the typical outcomes of foal-initiated communication, it follows that average suckling rate and average mare-foal distance did not predict future foal survival in our models. This pattern is at first counter intuitive, since suckling provides essential nutrition and proximity likely ensures foal safety. However, variation in these factors was minimal among foals (Table 3), suggesting strong stabilizing selection for ample suckling [59, 60] and mare-foal distances that allow for both maternal availability and environmental and social exploration [32, 53, 54]. Moreover, higher rates of mare-initiated communication did not correlate with average suckling rate, average mare-foal distance, or average mare condition during the time period studied, indicating two non-mutually exclusive possibilities: 1) that mare-initiated communication in and of itself may be important for foal survival; and/or 2) that mare-initiated communication predicted factors other than suckling, mare-foal distance, and mare condition that were important determinants of foal survival to nutritional independence.

The fact that foals of more communicative mares were more likely to survive indicates the importance of maternal "style" to offspring survival. Moreover, it suggests that, in feral horses, communication may provide a mechanism by which mothers facilitate both offspring independence and protection: without necessarily changing their foals' proximity or their exploration of the environment, mares can inform offspring of their (the mares') presence should they (the foals) need nutrients and/or protection. If true, it may be that mares assess current risks, both social and environmental, and use snorts to signal both their position and that it is "safe" for their foals to continue in their current activities. In this way, mares may modulate risk to their foals, again, ensuring foal safety while also encouraging independence. Less communicative mares may simply be less readily available to their offspring. Moreover, variation in how mares initiated communication in different ecological contexts indicates the importance of contact maintenance with their offspring. For example, in areas of lower visibility, mares were less likely to use snorts and more likely to use whinnies, perhaps to ensure information transfer and/or to better elicit foal responses. The fact that foals were less likely to alter their signaling according to the surrounding ecology may reflect a lack of experience that is only attained with additional development. For example, with increasing development, bottlenose dolphins (*Tursiops truncatus*) demonstrate vocal learning [61], chimpanzees (*Pan troglodytes*) exhibit increased gesturing proficiency [62], and treehoppers (*Umbonia crassicornis*) increase alarm thresholds [63].

Taken together, our results may shed some light on the role of the juvenile stage in mammals. Whether the behaviors exhibited during the juvenile period confer immediate or long-term benefits has long been a point of debate [8–12]. The behavioral flexibility exhibited by both mares and foals when initiating communication supports our hypothesis that current behavior is important to offspring's immediate circumstances. If the behaviors exhibited during the juvenile stage afforded only future benefits, one might expect them to be more static, to ensure that young mammals achieved specific milestones when necessary. Instead, we see more dynamic behaviors with mothers and/or offspring preferentially using signals associated with specific outcomes and responding to increasing offspring age, the surrounding ecology, and to each other. From the foals' perspective, this flexibility may ensure that they secure nutrients from and/or closer proximity to their mothers, providing support for the notion that immediate benefits are an important component of the behavior exhibited during the juvenile

stage. This is the case with several species [55], further suggesting that offspring behavior can afford immediate benefits during the juvenile period. On the other hand, if mares are in fact using communication as a means of modulating both social and environmental risks for their foals, mare behavior may serve a dual function: ensure not only that foals survive the hazards of the day, but also that critical interactions and experiences important to future development occur. For example, our data suggest that mares' communicative flexibility ensures that contact is maintained with their foals, should the immediate need for protection and/or nutrients arise. In addition, the association between mare-initiated communication early in development and foal survival to nutritional independence provides evidence that mare behavior exhibited during the juvenile stage can be critical for future benefits as well. Offspring allowed to play with conspecifics, interact with elders, and explore their environment are more likely to survive [32, 54, 64, 65]. It may be that more communicative mares are facilitating these experiences for their foals, contributing to their future survival [32]. Notably, the fact that mare behavior seems to be more critical than foal behavior regarding foal survival highlights the importance of studying the behavior of both mothers and offspring when examining the mammalian juvenile stage. Overall, our results suggest that appropriate responses to current circumstances (enabled by increased behavioral flexibility and adaptability) can help secure immediate benefits [33], but that the behavior demonstrated during the early juvenile stage can also be crucial to future survival [9].

## Supporting information

**S1 Table. Individual study subject details.** (-) indicate that mares were not observed in a particular year; (&) indicate double male bands; (*) indicate changes in band stallion within that year of observation.
(DOCX)

**S2 Table. AICc analysis for models with and without band as a random factor.**
(DOCX)

**S3 Table. Model output for insignificant main effects.**
(DOCX)

**S4 Table. Models separately estimating the effects of initiator type and signal use on the likelihood of suckling, distance decrease, and no-change outcomes.**
(DOCX)

**S1 File. Mare snort.**
(WAV)

**S2 File. Mare nicker.**
(WAV)

**S3 File. Mare whinny.**
(WAV)

## Acknowledgments

Thanks to James S. Adelman for his statistics help and his careful consideration of previous versions of this manuscript. Thanks also to Leanne Proops for providing the mare sound files and her assistance during revision. Special appreciation to Mañuel Vásquez and Rachel Gahan for their assistance in the field. Finally, we would like to dedicate this work to Dr. Henry Horn —his ability to find cool biology in even the most unexpected of places was an inspiration; this

work is the direct result of his dogged support and constant encouragement. He will be sorely missed.

## Author Contributions

**Conceptualization:** Cassandra M. V. Nuñez, Daniel I. Rubenstein.

**Data curation:** Cassandra M. V. Nuñez.

**Formal analysis:** Cassandra M. V. Nuñez.

**Funding acquisition:** Daniel I. Rubenstein.

**Investigation:** Cassandra M. V. Nuñez.

**Methodology:** Cassandra M. V. Nuñez, Daniel I. Rubenstein.

**Project administration:** Daniel I. Rubenstein.

**Resources:** Daniel I. Rubenstein.

**Supervision:** Daniel I. Rubenstein.

**Writing – original draft:** Cassandra M. V. Nuñez.

**Writing – review & editing:** Cassandra M. V. Nuñez, Daniel I. Rubenstein.

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
