## [Decision Letter · Decision Letter 0]

27 Feb 2020

PONE-D-19-24399

Communication is key: mother-offspring signaling can affect behavioral responses and offspring survival in feral horses (Equus caballus)

PLOS ONE

Dear Dr Nuñez,

Thank you for submitting your manuscript to PLOS ONE. After careful consideration, we feel that it has merit but does not fully meet PLOS ONE’s publication criteria as it currently stands. Therefore, we invite you to submit a revised version of the manuscript that addresses the points raised during the review process.

The reviewers and I enjoyed the manuscript, and the study provided novel insights into mare-foal relationships. The reviewers have suggested  revisions that should be addressed as they will improve the manuscript. They are outlined below.

We would appreciate receiving your revised manuscript by Apr 12 2020 11:59PM. To enhance the reproducibility of your results, we recommend that if applicable you deposit your laboratory protocols in protocols.io, where a protocol can be assigned its own identifier (DOI) such that it can be cited independently in the future. For instructions see: http://journals.plos.org/plosone/s/submission-guidelines#loc-laboratory-protocols

We look forward to receiving your revised manuscript.

Kind regards,

Elissa Z. Cameron

Academic Editor

PLOS ONE

Journal Requirements:

2. In your Methods section, please provide additional location information of the study area, including geographic coordinates for the data set if available.

Reviewers' comments:

Reviewer's Responses to Questions

**Comments to the Author**

1. Is the manuscript technically sound, and do the data support the conclusions?

Reviewer #1: Yes

Reviewer #2: Yes

2. Has the statistical analysis been performed appropriately and rigorously? 

Reviewer #1: Yes

Reviewer #2: Yes

3. Have the authors made all data underlying the findings in their manuscript fully available?

Reviewer #1: Yes

Reviewer #2: Yes

4. Is the manuscript presented in an intelligible fashion and written in standard English?

Reviewer #1: Yes

Reviewer #2: Yes

5. Review Comments to the Author

Reviewer #1: This study into mother-offspring vocal communication in feral horses aims to elucidate the function of mother-offspring signals and their ultimate benefits in terms of offspring survival. The authors use an extensive dataset to explore the context of a large number of signaling events in a population where foal mortality was significant, hence allowing exploration of the effects of mother-offspring communication on foal survival.

I believe this study offers significant insights into mother-offspring communication in horses. In particular, the result that, after controlling for mare-foal distance, mare body condition and suckling rates, a strong effect of communication rates on foal survival persists allows novel insights into the effect of maternal style on offspring survival in mammals. I therefore recommend this work be published in its current form with just a few minor recommended changes, as listed below.

Line 51, 301 – less/more frequently

Lines 80-81 – I would recommend adding here that this stable group composition allows for persistent social bonds between mares, citing Stanley et al 2018 (Stanley, C.R., Mettke-Hofmann, C., Hager, R. and Shultz, S., 2018. Social stability in semiferal ponies: networks show interannual stability alongside seasonal flexibility. Animal Behaviour, 136, pp.175-184)

Lines 92-3 – this is perhaps a little over-stated – does this study really allow insights to be gained into the function of the juvenile stage of mammals? I would argue instead that your study allows insights into the effects of maternal style on mammalian offspring survival. I would recommend more emphasis on maternal style here and in the discussion as I feel this more closely underpins your results.

Line 101 – should this be mare-foal distance increases?

Lines 102-3 – rephrase – not sure what is meant here

Lines 163, 282 – identify whose personal observation this was of the two authors

Lines 167-9 – I would argue that the type of snort used as an alarm call – typically a short, loud, harsh snort – is different in structure and function to the softer snort you identified in your study. The former would be given by a horse in a highly aroused state, with its head up, as an alarm call whereas the softer snort you describe in your study would be initiated by a horse in a more relaxed state as a contact call or sign of contentment. The second type is referred to as a sigh by Claudia Feh (Feh, C., 2005. Relationships and communication in socially. In The domestic horse: The origins, development and management of its behaviour, p.83). I suggest you refer to this as a sigh to be consistent with this text and to distinguish this from an alarm call snort.

Lines 192-3 – move this information to table legend and refer reader to table in text

Lines 239-40 – state somewhere how many mothers were repeat sampled in different years with different foals

Line 306 – interaction between

Lines 535-5 – I am glad you include this hypothesis as this reflects my initial thoughts on your results

Line 570 – mares’

Reviewer #2: Reviewer comment:

Generally a very thoughtful paper that has high relevance for bio-acoustic researchers and useful for a better understanding of the mother-offspring bond. The paper is well written and the aims are well formulated. These aims are achieved with this paper.

The major aspect that is missing is the lack of recent data of better quality to compare this unique dataset with. Indeed, technics and knowledge have evolve quite a lot since 1995 and it would have been interesting to include more recent data in order to compare the different call type with.

In the discussion section, I would suggest including the more recent publication on horses’ vocalisations. Indeed, at least two papers have been recently published on domestic horses as well as prewaski’s horses (the later also considered to be feral; Gaunitz, et al., 2018). I would thus suggest including the following papers: Briefer et al., 2015 and Maigrot et al., 2017.

Specific comments:

Abstract:

Line 24: “feral horse” does not seem to be precise enough. It would be interesting to include the breed or to give some more details as Przewalski’s horses are also considered to be feral, there is quite a wide variety of feral horses. Also on line 71, 76, 77, 82, 84, 115, 118, 120, 167, 178, 179, 442, 503, 504 and 537.

Introduction:

Line 57: The Latin name of mice (Mus musculus) is missing.

Line 77-81: It might be interesting to describe multi male bands (also called Bachelor groups) as well as it is mentioned on line 120.

Line 83: You reference only Tyler, 1972 and Feist et al., 1976. It might be relevant to include more recent publications as well (e.g. Waring, 2003 and Yeon, 2012).

Line 86-90: This part seems to already describe the beginning of the methods even though it is not yet the end of the methods. It might be relevant to only mention the aims and describe the data types later in the methods section.

Methods:

Line 138: The data have been recorded in 1995, 1996 and 1997. It would have been interesting to conduct another recording sessions more recently with more modern recording material in order to compare the old data to the new ones. It would add a lot to the article.

Line 166-169: Snorts are described as alarm signals in this section. However, snorts have also been described in play situations (McDonnell, 2003; Yeon, 3012) as well as when the nasal passage is irritated (Kiley, 1972) and when the horse is “restless and yet contained” (Waring, 2003). In addition, a really recent paper suggest snorts might indicate positive emotions (Stomp et al., 2018). It might thus be interesting to look at these alternative explanations to explain your founding and your data.

Results and Discussion:

Generally, the resultst and Discussion are really well written and well describe. There are no big suggestions.

6. PLOS authors have the option to publish the peer review history of their article (what does this mean?). If published, this will include your full peer review and any attached files.

Reviewer #1: Yes: Dr Christina R Stanley

Reviewer #2: No

---

## [Author Response · Author response to Decision Letter 0]

19 Mar 2020

PONE-D-19-24399

Communication is key: mother-offspring signaling can affect behavioral responses and offspring survival in feral horses (Equus caballus)

PLOS ONE

Dear Dr Nuñez,

Thank you for submitting your manuscript to PLOS ONE. After careful consideration, we feel that it has merit but does not fully meet PLOS ONE’s publication criteria as it currently stands. Therefore, we invite you to submit a revised version of the manuscript that addresses the points raised during the review process.

The reviewers and I enjoyed the manuscript, and the study provided novel insights into mare-foal relationships. The reviewers have suggested revisions that should be addressed as they will improve the manuscript. They are outlined below.

We would appreciate receiving your revised manuscript by Apr 12 2020 11:59PM. To enhance the reproducibility of your results, we recommend that if applicable you deposit your laboratory protocols in protocols.io, where a protocol can be assigned its own identifier (DOI) such that it can be cited independently in the future. For instructions see: http://journals.plos.org/plosone/s/submission-guidelines#loc-laboratory-protocols

• A rebuttal letter that responds to each point raised by the academic editor and reviewer(s). This letter should be uploaded as separate file and labeled 'Response to Reviewers'.

• A marked-up copy of your manuscript that highlights changes made to the original version. This file should be uploaded as separate file and labeled 'Revised Manuscript with Track Changes'.

• An unmarked version of your revised paper without tracked changes. This file should be uploaded as separate file and labeled 'Manuscript'.

We look forward to receiving your revised manuscript.

Kind regards,

Elissa Z. Cameron

Academic Editor

PLOS ONE

Journal Requirements:

We have made the necessary corrections to ensure that our manuscript meets PLOS ONE’s style requirements, including those regarding file naming. 

2. In your Methods section, please provide additional location information of the study area, including geographic coordinates for the data set if available.

We have added this information—please see lines 131-133.

We have added this information—please see lines 304-308. Due to the strictly observational nature of the research, neither the Princeton University nor the National Park Service IACUCs deemed permitting necessary. In addition, at the time of this study, the NPS did not require research permits of our research group—we were not required to submit research permits until 2013.

Thank you; we will make the data available upon our manuscript’s acceptance.

Reviewers' comments:

Reviewer's Responses to Questions

Comments to the Author

1. Is the manuscript technically sound, and do the data support the conclusions?

Reviewer #1: Yes

Reviewer #2: Yes

2. Has the statistical analysis been performed appropriately and rigorously?

Reviewer #1: Yes

Reviewer #2: Yes

3. Have the authors made all data underlying the findings in their manuscript fully available?

Reviewer #1: Yes

Reviewer #2: Yes

4. Is the manuscript presented in an intelligible fashion and written in standard English?

Reviewer #1: Yes

Reviewer #2: Yes

5. Review Comments to the Author

Reviewer #1: This study into mother-offspring vocal communication in feral horses aims to elucidate the function of mother-offspring signals and their ultimate benefits in terms of offspring survival. The authors use an extensive dataset to explore the context of a large number of signaling events in a population where foal mortality was significant, hence allowing exploration of the effects of mother-offspring communication on foal survival.

I believe this study offers significant insights into mother-offspring communication in horses. In particular, the result that, after controlling for mare-foal distance, mare body condition and suckling rates, a strong effect of communication rates on foal survival persists allows novel insights into the effect of maternal style on offspring survival in mammals. I therefore recommend this work be published in its current form with just a few minor recommended changes, as listed below.

Line 51, 301 – less/more frequently

Completed—thank you. Please see lines 58 and 337.

Lines 80-81 – I would recommend adding here that this stable group composition allows for persistent social bonds between mares, citing Stanley et al 2018 (Stanley, C.R., Mettke-Hofmann, C., Hager, R. and Shultz, S., 2018. Social stability in semiferal ponies: networks show interannual stability alongside seasonal flexibility. Animal Behaviour, 136, pp.175-184)

Added—thank you for this suggestion and for providing the reference. Please see lines 88-89.

Lines 92-3 – this is perhaps a little over-stated – does this study really allow insights to be gained into the function of the juvenile stage of mammals? I would argue instead that your study allows insights into the effects of maternal style on mammalian offspring survival. I would recommend more emphasis on maternal style here and in the discussion as I feel this more closely underpins your results.

Thank you for this comment. We definitely agree that our results give important information about maternal style and offspring survival. However, we also maintain that because our data include behavioral responses from both offspring and their mothers, our results have something important to say about the function of the juvenile stage in mammals; though as you outline, the insights gained are perhaps less “direct”. We have therefore: 1) added language about what our results have to say about the variation in and importance of maternal style to offspring survival and, 2) have softened our language both here and in the discussion.

Please see lines 99-103, 568-569, 587, 596, 603-604.

Line 101 – should this be mare-foal distance increases?

That line is as we intended—we predicted that communication would decrease and that decreases in mare-foal distance (so approaches by mares and/or foals towards one another) would decrease as foals grew older. Also, we have reworded this section to hopefully clarify our meaning (see our response to the next comment).

Lines 102-3 – rephrase – not sure what is meant here

Thank you for this comment—we apologize for any confusion. We have re-worded this section to read: 

We also predicted that with increasing foal age, 3) rates of mare-foal communication would decrease and that; 4) the outcomes of mare-foal communication would shift such that: a) decreases in mare-foal distance (approaches to one another) would decline in frequency; b) suckling behavior would decline in frequency; c) increases in mare-foal distance (leaves from one another) would increase in frequency; and d) no changes (to mare-foal distance or activity) would increase in frequency (please see lines 108-114).

We hope that this makes our predictions regarding the outcomes of mare-foal communication clearer. 

Lines 163, 282 – identify whose personal observation this was of the two authors

Completed—thank you. Please see lines 183 and 316.

Lines 167-9 – I would argue that the type of snort used as an alarm call – typically a short, loud, harsh snort – is different in structure and function to the softer snort you identified in your study. The former would be given by a horse in a highly aroused state, with its head up, as an alarm call whereas the softer snort you describe in your study would be initiated by a horse in a more relaxed state as a contact call or sign of contentment. The second type is referred to as a sigh by Claudia Feh (Feh, C., 2005. Relationships and communication in socially. In The domestic horse: The origins, development and management of its behaviour, p.83). I suggest you refer to this as a sigh to be consistent with this text and to distinguish this from an alarm call snort.

Thank you for this comment. Reviewer 2 also commented on this section, outlining how the snort may have several meanings/functions—we have added these alternate functions to the manuscript (please see lines 184-202). While we see your point regarding the “sigh”, we believe the signal we observed was more similar to that mentioned by McDonnell, Stomp, Waring, and Yeon [1-3], all of whom use the term “snort” when referring to this particular signal. Interestingly, in an edited version of Feh’s book chapter [4] she does refer to the “snort” and not the “sigh”. We have therefore elected to continue referring to this particular signal as a snort. We do hope that this is acceptable. 

Lines 192-3 – move this information to table legend and refer reader to table in text

Completed—please see lines 224-226.

Lines 239-40 – state somewhere how many mothers were repeat sampled in different years with different foals

We have added this information to the Study Subjects section (please see lines 165-166).

Line 306 – interaction between

Completed—thank you. Please see lines 342.

Lines 535-5 – I am glad you include this hypothesis as this reflects my initial thoughts on your results

Thank you! 

Line 570 – mares’

Completed—thanks for catching that!

Reviewer #2: Reviewer comment:

Generally a very thoughtful paper that has high relevance for bio-acoustic researchers and useful for a better understanding of the mother-offspring bond. The paper is well written and the aims are well formulated. These aims are achieved with this paper.

The major aspect that is missing is the lack of recent data of better quality to compare this unique dataset with. Indeed, technics and knowledge have evolve quite a lot since 1995 and it would have been interesting to include more recent data in order to compare the different call type with.

In the discussion section, I would suggest including the more recent publication on horses’ vocalisations. Indeed, at least two papers have been recently published on domestic horses as well as prewaski’s horses (the later also considered to be feral; Gaunitz, et al., 2018). I would thus suggest including the following papers: Briefer et al., 2015 and Maigrot et al., 2017.

Thank you for this suggestion and these great references—the work is really very interesting. We now refer to these studies in line 507.

Specific comments:

Abstract:

Line 24: “feral horse” does not seem to be precise enough. It would be interesting to include the breed or to give some more details as Przewalski’s horses are also considered to be feral, there is quite a wide variety of feral horses. Also on line 71, 76, 77, 82, 84, 115, 118, 120, 167, 178, 179, 442, 503, 504 and 537.

Thank you for this comment. We have added information about the Shackelford horses’ origin to the Study Area section (please see lines 134-137). We have explained that they are classified as Colonial Spanish horses, that they are similar to South American Iberian breeds, and are most closely related to the Venezuelan Criollo, Puerto Rican Paso Fino, and Marsh Tacky horses. We have left the term “feral horse” throughout the manuscript, however, because the equids of Shackleford Banks 1) are not referred to as one particular breed (they are closely related to several); 2) have been referred to as feral horses and their genus and species as (Equus caballus) for several decades [5, 6] and; 3) are defined as such by the National Park Service (https://www.nps.gov/calo/learn/nature/horse-faqs.htm), the agency managing the herd. We hope that this is acceptable.

Introduction:

Line 57: The Latin name of mice (Mus musculus) is missing.

We did not designate the mice used in these experiments as Mus musculus as the authors used a specific lab strain of mice and did not use this designation. That said, most lab mice are of Mus musculus so we have made the addition (please see lines 64-65).

Line 77-81: It might be interesting to describe multi male bands (also called Bachelor groups) as well as it is mentioned on line 120.

Thank you for this comment. We apologize for the confusion. When we used the term multi-male bands, we were referring to bands with more than one stallion, their females, and their offspring, not to bachelor groups. We see how this term could have been confusing and so have changed “multi-male” to “multi-stallion” as used in Linklater [7] (please see line 139). We prefer not to add a description of bachelor groups as they were not a focus of our study.

Line 83: You reference only Tyler, 1972 and Feist et al., 1976. It might be relevant to include more recent publications as well (e.g. Waring, 2003 and Yeon, 2012).

Thank you for this suggestion—we have added these references (please see lines 91).

Line 86-90: This part seems to already describe the beginning of the methods even though it is not yet the end of the methods. It might be relevant to only mention the aims and describe the data types later in the methods section.

Thank you for this comment. We can see your point, however, we prefer to keep the language as stated. The current format gives the reader general information about how we used field data to address the questions in our study earlier rather than later—we feel that this prepares the reader for what is to come later. We hope that is acceptable.

Methods:

Line 138: The data have been recorded in 1995, 1996 and 1997. It would have been interesting to conduct another recording sessions more recently with more modern recording material in order to compare the old data to the new ones. It would add a lot to the article.

We apologize, but we are not quite certain about what you are referring to when you say “recording sessions”. If you are referring to our behavioral data collection, the addition of unfortunately, another field season is not possible at this point given our current responsibilities/commitments. In addition, we maintain that three years of data collection, regardless of when they occurred, constitute a significant dataset.

On the other hand, you may be referring to the recorded signals (snort, nicker, and whinny) and their spectrograms. In fact, we did not make recordings of the mare/foal calls during this study. The sound files provided (actually in response to a previous reviewer’s suggestion) were recorded by Leanne Proops and are from Misaki mares with foals—they are meant to provide the reader with context regarding the nature of the different signals; as such, these recordings in and of themselves are not the focus of the manuscript. We originally cited Dr. Proops in the manuscript but were advised by a previous reviewer to delete that reference. We now include Dr. Proops in our Acknowledgements where we thank her for the recordings (please see original lines 586-587; now new lines 619-620). We are more than happy to reference her within the manuscript if the Editor feels that is the best course of action.

Line 166-169: Snorts are described as alarm signals in this section. However, snorts have also been described in play situations (McDonnell, 2003; Yeon, 3012) as well as when the nasal passage is irritated (Kiley, 1972) and when the horse is “restless and yet contained” (Waring, 2003). In addition, a really recent paper suggest snorts might indicate positive emotions (Stomp et al., 2018). It might thus be interesting to look at these alternative explanations to explain your founding and your data.

Thank you for this comment and for all of these references! We have added several of them (please see lines 186-202), though we did elect to leave out Kiley (1972), as clearing the nasal passages when irritated likely doesn’t qualify as meaningful communication, and also McDonnell (2003), as we were not able to gain access to that reference. While we recognize that mare and foal snorts may have certainly had more than one function, we maintain that our data suggest that they were primarily used as contact calls and have focused our interpretation of the results as such. Thank you again for this great comment and for providing these great references. 

Results and Discussion:

Generally, the resultst and Discussion are really well written and well describe. There are no big suggestions.

Thank you so much!

References

1. Stomp M, Leroux M, Cellier M, Henry S, Lemasson A, Hausberger M. An unexpected acoustic indicator of positive emotions in horses. PloS one. 2018;13(7):e0197898-e.

2. Waring GH. Horse behavior. Second ed. Norwich, NY, USA: Noyes Publications/William Andrew Publishing; 2003. 442 p.

3. Yeon SC. Acoustic communication in the domestic horse (Equus caballus). Journal of Veterinary Behavior. 2012;7(3):179-85.

4. Feh C. Relationships and communication in socially natural horse herds. The Domestic Horse: The Origins, Development and Management of Its Behaviour: Cambridge University Press; 2005. p. 83-93.

5. Conant E, Juras R, Cothran E. A microsatellite analysis of five Colonial Spanish horse populations of the southeastern United States. Animal Genetics. 2012;43(1):53-62.

6. Rubenstein DI. Behavioural ecology of island feral horses. Equine Veterinary Journal. 1981;13(1):27-34.

7. Linklater WL, Cameron EZ, Stafford KJ, Veltman CJ. Social and spatial structure and range use by Kaimanawa wild horses (Equus caballus: Equidae). New Zealand Journal of Ecology. 2000;24(2):139-52.

---

## [Editor Report · Decision Letter 1]

23 Mar 2020

Communication is key: Mother-offspring signaling can affect behavioral responses and offspring survival in feral horses (Equus caballus)

PONE-D-19-24399R1

Dear Dr. Nuñez,

We are pleased to inform you that your manuscript has been judged scientifically suitable for publication and will be formally accepted for publication once it complies with all outstanding technical requirements.

With kind regards,

Elissa Z. Cameron

Academic Editor

PLOS ONE
---

## [Editor Report · Acceptance letter]

6 Apr 2020

PONE-D-19-24399R1 

Communication is key: Mother-offspring signaling can affect behavioral responses and offspring survival in feral horses (*Equus caballus*) 

Dear Dr. Nuñez:

I am pleased to inform you that your manuscript has been deemed suitable for publication in PLOS ONE. Congratulations! Your manuscript is now with our production department. 

With kind regards,

on behalf of

Prof Elissa Z. Cameron 

Academic Editor

PLOS ONE